# Bootstrapping Reinforcement Learning with Imitation for Vision-Based Agile Flight

**Jiaxu Xing, Angel Romero, Leonard Bauersfeld, Davide Scaramuzza**
Robotics and Perception Group
University of Zurich, Switzerland
`{jixing, roagui, bauersfeld, sdavide}@ifi.uzh.ch`

**Abstract:** Learning visuomotor policies for agile quadrotor flight presents significant difficulties, primarily from inefficient policy exploration caused by high-dimensional visual inputs and the need for precise and low-latency control. To address these challenges, we propose a novel approach that combines the performance of Reinforcement Learning (RL) and the sample efficiency of Imitation Learning (IL) in the task of vision-based autonomous drone racing. While RL provides a framework for learning high-performance controllers through trial and error, it faces challenges with sample efficiency and computational demands due to the high dimensionality of visual inputs. Conversely, IL efficiently learns from visual expert demonstrations, but it remains limited by the expert's performance and state distribution. To overcome these limitations, our policy learning framework integrates the strengths of both approaches. Our framework contains three phases: training a teacher policy using RL with privileged state information, distilling it into a student policy via IL, and adaptive fine-tuning via RL. Testing in both simulated and real-world scenarios shows our approach can not only learn in scenarios where RL from scratch fails but also outperforms existing IL methods in both robustness and performance, successfully navigating a quadrotor through a race course using only visual information. Videos of the experiments are available at `https://bootstrap-rl-with-il.github.io/`.

**Keywords:** Quadrotor, Visuomotor Control, Reinforcement Learning

## 1 Introduction

Visuomotor policy learning enables robots to perform complex tasks by directly mapping visual information into action. This technique has been successfully demonstrated in various robotic systems to learn complex behaviors such as visual navigation [1, 2], dexterous manipulation [3, 4, 5], and agile maneuvers [6, 7, 8]. The ability to learn directly from visual data allows machines to interpret visual observations and translate them directly into corresponding motor actions, akin to the human skill of hand-eye coordination. However, learning from only visual inputs introduces a range of distinct challenges. The intrinsic high dimensionality of visual input makes the policy exploration and learning process more inefficient than using low-dimensional input, such as robot states.

In agile quadrotor flight, these challenges are more pronounced due to the platform's agility, inherent instability, and reliance on low-level commands like collective thrust and body rates, which necessitate precise low-latency closed-loop control. Previous works [8, 1] have demonstrated the capability of piloting quadrotors at high speeds using visual inputs for acrobatics and obstacle avoidance. These systems always rely on high-frequency proprioceptive data from an Inertial Measurement Unit (IMU) for accurate state estimation. However, despite these advancements, the goal of training and deploying an autonomous drone directly using RGB images has yet to be achieved. This limitation is particularly relevant in scenarios such as first-person-view (FPV) drone racing, where pilots achieve high-performance flight near the physical limits of the platform using only visual information from

8th Conference on Robot Learning (CoRL 2024), Munich, Germany.

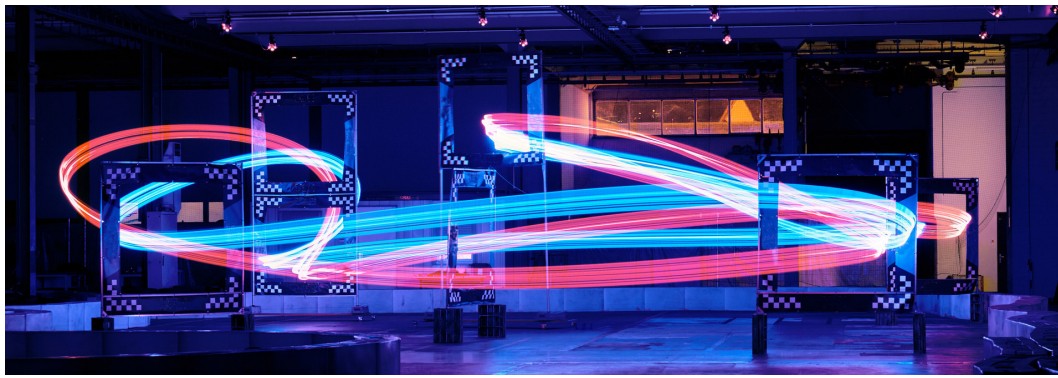

Figure 1: Long exposure image of real-world flights shows a blue trajectory for our approach and a red one for the imitation policy. Training on the same number of samples, our approach yields a tighter trajectory, resulting in faster lap times and demonstrating superior performance and robustness.

a 60 Hz monocular camera. Addressing this gap, our work specifically focuses on autonomous quadrotor racing, employing purely visual inputs without relying on metric state estimation or IMU data. Two primary learning methods have emerged in tackling this challenge: Reinforcement Learning (RL) and Imitation Learning (IL). RL is gaining traction as a general framework for designing complex controllers that are difficult to handcraft using classical methods [9, 7, 10]. This method involves learning through millions of trial-and-error interactions within a simulated environment. Despite the potential of RL, it often requires collecting extensive data samples to learn effectively, which can be computationally demanding. As a result, enhancing the sample efficiency of RL algorithms has become a critical focus. Thus, the ability to explore and learn *efficiently* in vision-based RL environments from scratch is essential, presenting a key challenge that our research seeks to overcome.

Conversely, IL has been successfully demonstrated on different mobile platforms to perform end-to-end vision-based learning [8, 1, 11, 5]. IL synthesizes policies using either expert demonstrations or a privileged policy in a supervised fashion. The optimization objective is to clone an expert rather than to interact with the environment and collect rewards as in RL. Due to this simplification, IL typically requires fewer samples and, consequently, has been validated for real-world robot learning problems [1, 8, 5, 11, 12, 13]. However, IL faces several challenges, including the significant issue of *covariate shift*. This term describes the discrepancies between training data (demonstrations for IL) and real-world scenarios, which can lead to degraded performance as the learned policy may not generalize well to new situations. Additionally, the effectiveness of an IL-derived policy is inherently limited to the quality of the expert demonstrations it is based on; it cannot surpass the performance of the demonstrations used for training.

The RL policy for state-of-the-art autonomous drone racing [7], which outperformed world-champion pilots, still relies on explicit state estimates, including position, velocity, and orientation. Our work, however, focuses on learning visuomotor policies that map visual information directly to control commands without explicit state estimation. Achieving this would bring autonomous flying machines closer to how human pilots navigate. However, this ambition was unattained in the realm of drone racing due to one fundamental challenge: *sample inefficiency*.

**Contributions** By leveraging the complementary advantages of IL and RL, we propose a framework that trains a policy capable of navigating through a sequence of gates using solely gate corners or RGB images. Through experiments in both simulation and real-world environments, we demonstrate that our approach, given the same sample budget, outperforms existing IL methods in robustness and performance and succeeds where RL from scratch fails. While most state-of-the-art visuomotor policy learning for mobile robotics adopts the teacher-student IL framework [8, 1, 6, 14], we extend this framework by bootstrapping the IL policy for adaptive RL fine-tuning to enhance the performance. Although we validate our method using vision-based drone racing, our approach does not rely on task-specific adaptations that might limit its applicability to other robotic platforms or tasks.

## 2 Related Work

**Vision-Based Robot Learning** Deep visuomotor policies directly map actions from visual inputs, such as RGB images [4] or depth images [1, 15]. In contrast to conventional methods, end-to-end approaches often operate without the need of environmental mapping [16, 17], precise state estimation [18, 19], or motion planning [18, 20]. Existing works have demonstrated end-to-end vision-based policies primarily through either RL or IL [8, 21, 1, 22, 23]. In RL, the optimization objective is shaped by task-specific reward designs. The policy gains robustness and generalizability by learning from both positive and negative samples through interactions with training environments [24]. Despite having a task-centric objective, vision-based RL has predominantly found application in simulations, such as fixed-based manipulation tasks [25, 21], or video games [26], due to its sample inefficiency. Therefore, efficient approaches for vision-based RL in mobile robots are still actively being explored. In contrast, in most IL approaches, the optimization objective is a difference measure in behavior between the learned policy and the expert demonstrations [27, 28]. The simplification of learning the task solely from expert demonstrations dramatically reduces the required samples for effective training. Thanks to its sample efficiency, various visuomotor policy learning methods have been demonstrated on mobile robots [8, 1, 6, 11, 5]. Due to the supervised fashion of policy training, the performance of IL policies is capped by the teacher policies and suffers degradation for out-of-distribution observations [28], particularly in task settings where actions are only partially observable from visual inputs.

**RL Finetuning from Expert Demonstrations** For various robot learning applications, one central question is how expert demonstrations can be used to develop high-performance policies. The predominant method in existing research begins with behavior cloning (BC) and refines the policy through RL to enhance state exploration and sample efficiency, as seen in tasks like pole balancing and dexterous manipulation [29, 30, 31, 32, 33]. A major challenge in online fine-tuning with RL is the "stability-plasticity dilemma", where networks catastrophically forget old behaviors while continuing learning [34]. This problem is especially acute for RL settings, as exploration generates out-of-distribution (OOD) data, leading to suboptimal learning or even completely unlearning the previous skills. Using our approach, which adaptively updates the policy based on performance, we demonstrated that catastrophic forgetting is greatly reduced.

## 3 Methodology

The drone racing task can be formulated as an optimization problem where the objective is to minimize the time required to navigate through a predefined sequence of gates [35], as illustrated in Fig. 4. The drone perceives the environment solely through a single RGB camera, and the learned policy network utilizes egocentric vision input $o_p$ to output Collective Thrust and Bodyrates control

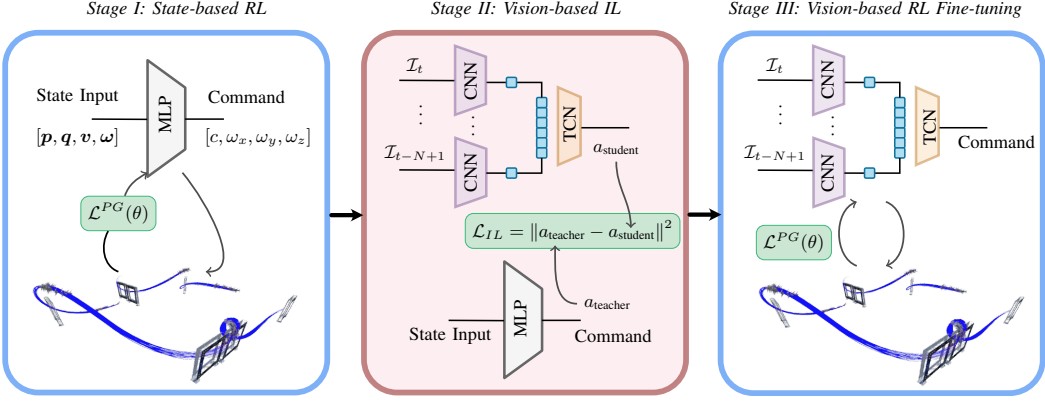

Figure 2: We demonstrate visuomotor policy learning in three different stages. In stage I, we train a state-based teacher policy using RL. In stage II, we use IL to learn a student distillation policy using visual inputs. In stage III, we bootstrap the actor using the student policy to fine-tune the policy through vision-based RL.

command $[c, \omega_x, \omega_y, \omega_z]$, where $c$ represents collective thrust, and $\omega$ denotes the body rates [36]. As shown in Fig. 2, our approach consists of three phases: (I) initial training of a teacher policy using state information, (II) distillation into a student policy via IL to transfer knowledge and create an efficient baseline model, and (III) fine-tuning the student policy through a novel performance-constrained adaptive RL approach to enhance the policy's performance and robustness.

**Phase I: State-based Teacher Policy Training** The teacher policy $\pi_{\text{teacher}}$ processes state observations $\boldsymbol{s} = \left[\boldsymbol{p}, \tilde{\boldsymbol{R}}, \boldsymbol{v}, \boldsymbol{\omega}, \boldsymbol{i}, \boldsymbol{d}\right]$, where $\boldsymbol{p} \in \mathbb{R}^3$ denotes the drone's position, $\tilde{\boldsymbol{R}} \in \mathbb{R}^6$ is a vector comprising the first two columns of $\boldsymbol{R}_{\mathcal{WB}}$ [37], $\boldsymbol{v} \in \mathbb{R}^3$ and $\boldsymbol{\omega} \in \mathbb{R}^3$ denote the linear and angular velocity of the drone, and $\boldsymbol{d} \in \mathbb{R}^3$ represents the position of the next gate center relative to the current drone position. The training of the teacher policy employs a model-free reinforcement learning approach using Proximal Policy Gradient (PPO) [24]. The RL policy training rewards are adjusted based on [38]. The reward at time $t$, denoted as $r_t$, is defined as the sum of various components

$$r_t = r_t^{\text{prog}} + r_t^{\text{perc}} + r_t^{\text{act}} + r_t^{\text{br}} + r_t^{\text{pass}} + r_t^{\text{crash}}, \tag{1}$$

where $r_t^{\text{prog}}$ encourages progress towards the next gate to be passed [39], $r_t^{\text{perc}}$ encodes perception awareness by adjusting the quadrotor's attitude such that the optical axis of its camera points towards the next gate's center, $r_t^{\text{act}}$ penalizes action changes from the last time step, $r_t^{\text{br}}$ penalizes bodyrates and consequently reduces motion blur, $r_t^{\text{pass}}$ is a binary reward that is active when the robot successfully passes the next gate, $r_t^{\text{crash}}$ is a binary penalty that is only active when a collision happens, which also ends the episode. For the detailed reward formulation, we refer the readers to the Appendix.

**Phase II: Imitation Learning using Visual Input** The goal of this phase is to distill a student policy $\pi_{\text{student}}$ from the expert $\pi_{\text{teacher}}$. The vision-based student policy takes a sequence (history length $H$ timesteps) of perceptual observations $[\boldsymbol{o}_{t-H+1}, \ldots, \boldsymbol{o}_t]$ as input. We use a Temporal Convolutional Network (TCN) [40] to encode the series of vision embeddings from $\mathcal{I}$ or corners $\mathcal{C}$. The output features from the two TCNs are concatenated and fed into a two-layer MLP, which outputs the actions. The supervision loss is formulated as $\mathcal{L}_A$ (action error), which is the mean square error between the outputs of the teacher policy and the student policy

$$\mathcal{L}_A(\mathcal{D}, \theta_{\text{student}}) = \mathbb{E}_{\mathcal{D}}\left[\|\pi_{\text{student}}([\boldsymbol{o}_{t-N+1}, \ldots, \boldsymbol{o}_t]; \theta_{\text{student}}) - \pi_{\text{teacher}}(\boldsymbol{s}_t)\|\right]. \tag{2}$$

To acquire an imitation learning policy, the most common methods are (i) Behavior Cloning (BC) or (ii) DAgger [27]. In the case of BC, the state-based teacher policy is executed for a fixed number of steps, generating a dataset that encompasses corresponding perceptual observations and action outputs. In contrast, DAgger involves iteratively training the student policy by executing the learned student policy with a gradually increasing probability over time. The on-policy training fashion of DAgger results in a broader distribution of demonstrations from the student policy. This approach results in improved performance compared to traditional BC, contributing to more effective learning.

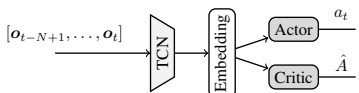

(a) Symmetric Actor-Critic configuration.

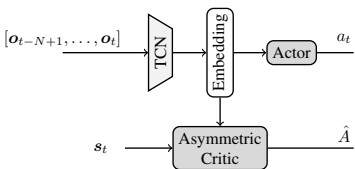

(b) Asymmetric Actor-Critic configuration.

Figure 3: Visualization of difference between the symmetric and asymmetric actor-critic learning setup.

**Phase III: Performance-Adaptive Online Fine-Tuning** State-of-the-art visuomotor policy learning for mobile robotics typically employs a teacher-student IL framework; however, integrating online RL fine-tuning could further enhance policy performance and robustness. The major challenge of online policy fine-tuning is catastrophic forgetting during exploration, where the state distribution shift during exploration causes the pre-trained policy to generalize poorly. To address this, we propose an algorithm that conditions exploration and network updates on the policy's performance, as shown in Algorithm 1. In our algorithm, we initiate the fine-tuning process using the pre-trained student policy as the starting actor in RL. However, a straightforward plug-and-play approach may not yield optimal results due to two key challenges:

(i) the critic function requires interactions to adapt the pre-trained actor, necessitating a "warm-up" process. (ii) During the initial training phase, policy update steps should be kept relatively adaptive based on the learning progress to prevent catastrophic forgetting resulting from sudden, large updates. In our approach, if the policy improvement is small, the update rate of the critic is automatically increased. Once the policy achieves high-reward action sequences, the policy update rate also increases. By linking the learning rates and the clip range to the policy rollout performance, we eliminate the need for heuristic tuning of learning rates to mitigate catastrophic forgetting. We believe this approach is easily generalizable to other platforms as it does not require task-specific information.

In our approach, the exploration and learning rates should dynamically depend on the agents' performance rather than being solely determined by the number of iterations.

---

**Algorithm 1** Adaptive Fine-tuning and Proximal Policy Gradient Update

---

1: **Input:** Pre-trained policy $\pi_{\text{pre}}$, learning rates $LR_\pi, LR_V$, constants $c_V, c_\pi, c_\epsilon$
2: Execute $\pi_{\text{pre}}$ to collect initial reward $r_{\text{init}}$
3: Initialize learning rates $LR_\pi$ for policy and $LR_V$ for value function
4: Freeze actor and exclusively train critic for $N$ iterations
5: **for** each training step **do**
6:     Sample batch of transitions $(s, a, r, s')$ from experience buffer
7:     Compute advantage estimates $\hat{A}$ and update critic $\theta \leftarrow \theta + LR_V \nabla_\theta \mathcal{L}_V(\theta)$
8:     **if** (step mod $N$) $== 0$ **then**
9:         Evaluate policy to collect reward $r_{\text{rollout}}$
10:        Compute performance ratio $\alpha = \frac{r_{\text{rollout}}}{r_{\text{init}}}$
11:        Update learning rates and PPO clip range:
12:        $LR_\pi \leftarrow \min\left(LR_\pi + \max(\alpha - 1, 0) \cdot c_V, \ LR_{\pi\text{max}}\right)$
13:        $LR_V \leftarrow \max\left(LR_V - \max(\alpha - 1, 0) \cdot c_\pi, \ LR_{V\text{min}}\right)$
14:        $\epsilon \leftarrow \min\left(\epsilon + \max(\alpha - 1, 0) \cdot c_\epsilon, \ \epsilon_{\text{max}}\right)$
15:     **end if**
16:     Update policy by maximizing PPO objective
17:     $\mathcal{L}_\pi(\phi) = \mathbb{E}\left[\min\left(\frac{\pi_\phi(a|s)}{\pi_{\text{old}}(a|s)}\hat{A}, \ \text{clip}\left(\frac{\pi_\phi(a|s)}{\pi_{\text{old}}(a|s)}, 1 - \epsilon, 1 + \epsilon\right)\hat{A}\right)\right]$
18:     $\phi \leftarrow \phi + LR_\pi \nabla_\phi \mathcal{L}_\pi(\phi)$
19: **end for**
20: **Output:** Policy parameters

---

**Asymmetric Critic** For the third phase vision-based RL, we employ an asymmetric actor-critic setup, as illustrated in Fig. 3 (b). This approach involves augmenting the critic function inputs with privileged information, such as the robot state $s$. Incorporating this privileged state information enables more precise learning of the critic function, thereby enhancing the efficiency and performance of actor training in partially observable settings [21].

## 4 Experiments and Results

### 4.1 Training Setup

We validate our approach on three different race tracks, as shown in Fig. 4, namely the *"SplitS"*, *"Figure 8"*, and *"Kidney"* tracks. During evaluation, each policy is rolled out until it completes 10 laps or for 3000 simulation time steps (50 seconds). We evaluate each policy 100 times, starting from positions uniformly sampled within a 1m cubic box centered on the nominal starting position of the racing track. Each method is repeated 5 times with different random seeds. The evaluation uses a realistic simulation based on the BEM model for aerodynamic effects [41].

We use three evaluation metrics: success rate (SR), mean-gate-passing-error (MGE), and lap time (LT). SR is the ratio of completed laps to total trials. MGE measures the distance between the drone's position and the gate center when passing through, here the inner gate size used for experiments is 1.5 m. LT indicates the duration to complete a full race track, flying through all gates. Further details on training configurations and our hardware setup are available in the Appendix.

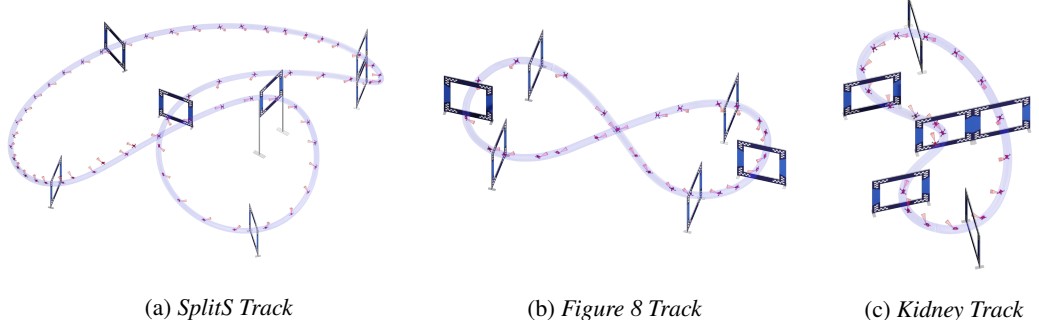

| (a) *SplitS Track* | (b) *Figure 8 Track* | (c) *Kidney Track* |

Figure 4: Visualization of the drone racing tracks used for the experiments, each characterized by varying levels of complexity. All the tracks maintain a consistent size scale, spanning widths from 8 meters to 16 meters.

**Policy Input** To evaluate the performance of our approach, we conduct IL experiments with two categories of visual input: (i) implicitly learned from ResNet [42]. Here we employ a pre-trained ResNet50 [42] network on ImageNet [43] to produce embeddings of length 128. Hence, the input of the TCN network corresponds to the history of the embeddings, which is $H \times 128$. (ii) racing gate corners projected in the camera pixel frame. Here we utilize the *normalized pixel coordinates* of all gate corners projected in the camera view. We model gate corners based on the gate detection system presented in [7]. To simulate real-world scenarios, we include domain randomization such as gate scales, pixel position noise (10 pixels in both $(u, v)$ in a $1280 \times 760$ image frame), and a 10% random missing probability for each corner at each frame to mimic the detection failures of the gate detector. After projection, we sort the gate corner pixels by their $u$ coordinates, and we model corners out of the image view as $(-1, -1)$. Detailed simulation visualizations for both inputs are available in the Appendix.

## 4.2 Experiment Results

**Performance comparison to baseline approaches** One inherent limitation of the student-teacher IL framework is to infer reasonable actions from partial information. This limitation arises because the student policy is trained only on the explicit actions of the expert, without understanding the underlying context that the expert may infer from unobservable cues. As a result, the student policy can struggle in scenarios where critical information is missing, leading to suboptimal actions and reduced overall performance. An example of this is the *SplitS Track*, where frames often lack visible corners (detailed in Appendix). Hence, to benchmark the learned policies' performance, we conduct a detailed analysis of our approach to the existing baselines using various time horizons for the policy $H \in [2, 4, 8, 16, 32, 64]$.

Table 1: Policy performance evaluation averaged by 6 different history lengths using both implicitly learned representations, specifically a common ResNet50 [42] for RGB images, and the task-specific gate corners positions across three different racing tracks.

| Input | Methods | \multicolumn{9}{c}{Race Tracks} | | | | | | | | |
|-------|---------|------|---------|-------|------|---------|-------|------|---------|-------|
| | | \multicolumn{3}{c}{*Figure 8*} | | | \multicolumn{3}{c}{*SplitS*} | | | \multicolumn{3}{c}{*Kidney*} | | |
| | | SR% | MGE [m] | LT [s] | SR% | MGE [m] | LT [s] | SR% | MGE [m] | LT [s] |
| | BC | 0 | crash | - | 0 | crash | - | 0 | crash | - |
| | RL [24] | 0 | crash | - | 0 | crash | - | 0 | crash | - |
| Pixel | DAgger [27] | 63 | 0.41 | 5.19 | 57 | 0.43 | 8.30 | 64 | 0.28 | 5.53 |
| | PIRLNav [22] | 21 | 0.49 | 5.79 | 4 | 0.67 | 8.32 | 18 | 0.61 | 6.30 |
| | **Ours** | **85** | **0.32** | **4.57** | **76** | **0.25** | **7.77** | **82** | **0.21** | **4.93** |
| | BC | 0 | crash | - | 0 | crash | - | 0 | crash | - |
| | RL [24] | 0 | crash | - | 0 | crash | - | 0 | crash | - |
| Corners | DAgger [27] | 57 | 0.42 | 4.82 | 52 | 0.51 | 8.81 | 60 | 0.34 | 5.42 |
| | PIRLNav [22] | 12 | 0.57 | 6.12 | 0 | crash | - | 12 | 0.69 | 6.48 |
| | **Ours** | **79** | **0.37** | **4.84** | **72** | **0.32** | **7.91** | **85** | **0.15** | **5.18** |

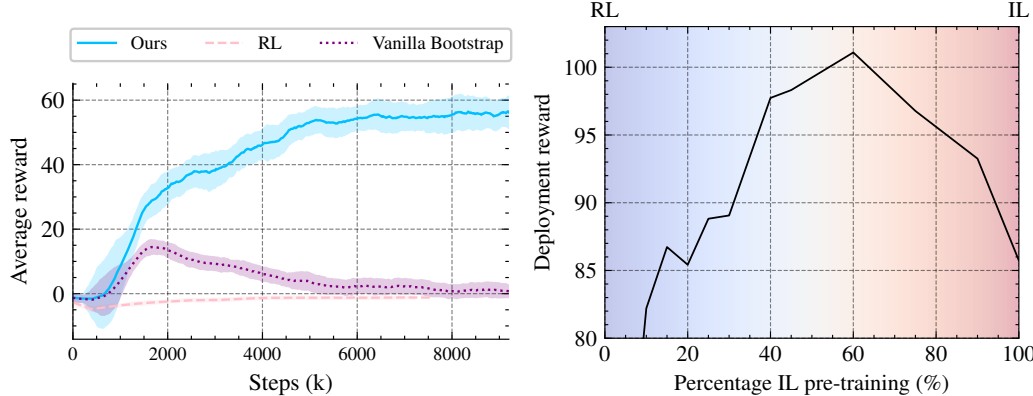

Figure 5: **Left:** Reward comparison between our approach and the other RL configurations. Ours is the only approach that is able to learn to perform the task. **Right:** Using a fixed sample budget, we varied the IL policy percentage for our approach on the vision-based policy for SplitS track, achieving a local maximum in deployment reward with 60% IL pretraining and 40% RL fine-tuning.

We included 4 baseline methods, namely BC and IL, RL from scratch, and a fine-tuning baseline, PIRLNav [22], where BC policy is fine-tuned using RL without adaptivity. We average the evaluation metrics, SR, MGE, and LT among various $H$. The results are shown in Table. 1. Across all configurations and racing tracks, our approach consistently exhibits the best performance in all the metrics compared to all baselines, under the same 10M data sample budget. In our approach, since our third stage takes advantage of an asymmetric critic containing privileged state information, the resulting policy will possess more task understanding. For the detailed metrics for the individual history length, we refer the readers to the Appendix.

**Training Effectiveness with different RL configurations** To demonstrate the effectiveness of our visuomotor policy learning approach, we ablate the training performance of our approach with different baseline RL configurations: (i) RL from scratch using the asymmetric critic with privileged states, and (ii) Vanilla fine-tuning, where we initialize the parameters of the actor network using the same pre-trained DAgger policy perform same RL training setup as used in stage I. This setup is distinct from the baseline PIRLNav where a BC policy is used for initialization. For (i) we train the RL policy using RGB images with 10M samples and our approach and baseline (ii) we use 5M data samples for pretraining and 5M data samples for fine-tuning. For a fair comparison, we selected the policy with $H = 32$ for all the methods, which balances good policy performance with the network's capability for real-time, low-latency control of our quadrotor. The results are illustrated in Fig. 5 left. Firstly, it is noteworthy that the direct RL from corners or pixels achieves a 0% success rate in all three tracks. This once again underscores the difficulty of RL exploration in high-dimensional time series without bootstrapping. The policy initially showed improved learning performance for the vanilla bootstrapping baseline but deteriorated with increasing timesteps. This highlights the importance of the adaptive component in our third phase for maintaining policy performance.

**Sample Efficiency Analysis** To analyze we fix the total sample budget at 10M and varying the pre-training ratio, we visualize the deployment reward of the resulting policies, shown in Fig. 5 right. It is evident that at 60%, the collected rewards achieve a peak (>100) in the curve representing the best performance. The plot indicates that by using an appropriate amount of data, the performance can easily surpass policies trained solely from IL. The detailed training metrics of individual policies are presented in the Appendix.

Table 2: Evaluation results on Perceptual and Positional disturbance.

| Disturbance | Prob. [%] | SR% IL | SR% Ours | Error [m] IL | Error [m] Ours |
|---|---|---|---|---|---|
| Perceptual | 1 | 59 | **100** | 0.38 | **0.25** |
|  | 5 | 33 | **91** | 0.55 | **0.39** |
| Positional | 1 | 84 | **100** | 0.32 | **0.28** |
|  | 5 | 64 | **97** | 0.49 | **0.33** |

**Robutstness to unknown disturbances** In the following experiments, we conduct two trials to verify whether our approach enhances the policy's robustness to unknown disturbances. To simulate real-world uncertainties, we conducted two experiments: i) random frame blackouts to mimic sensor failures like communication loss, and ii) random positional jumps during flight to simulate disturbances such as strong winds. The results are demonstrated in Table. 2, it is clear that our approach outperforms the baseline DAgger approach in terms of robustness to unknown disturbance.

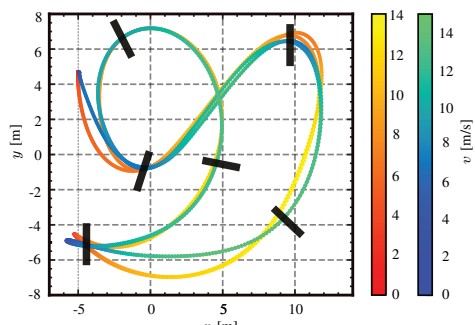

Figure 6: Comparison of real-world trajectories on the SplitS track: ours (green) achieves 7.67s lap time, while DAgger's (orange) is 8.06s.

**Towards the Limit of End-to-End Policy.** To explore how far we can push or approach achieving an end-to-end policy that maximizes the potential of the physical platform, we conducted an experiment aimed at pushing the speed to its limit. We applied our approach to train on the *SplitS* track, based on the setup described in [7], where the policy outperformed several human champions. In [7], the policy achieved a median lap time of 5.52 seconds, while the best human pilots achieved 5.76 seconds in the real world. The main difference from our previous experiments is that we increased the maximum thrust limit. We tested our approach using the aforementioned corner observation and validated it through 100 runs as part of our evaluation. The policy's performance resulted in an average lap time of 5.82 seconds, with the best policy achieving a lap time of 5.54 seconds. This indicates that an end-to-end policy's extreme performance can match that of human world champions.

**Real-world Experiments** To demonstrate policy improvements, we validated our policy in real-world scenarios using Hardware-in-the-Loop (HIL) simulations, aided by a VICON motion capture system for perceptual inputs. We compared our approach against

| Input | Methods | Race Tracks | | | | | |
| | | *Figure 8* | | *SplitS* | | *Kidney* | |
| | | MGE [m] | LT [s] | MGE [m] | LT [s] | MGE [m] | LT [s] |
|---|---|---|---|---|---|---|---|
| Corners | IL | 0.49 | 4.85 | 0.55 | 8.18 | 0.37 | 5.76 |
| | **Ours** | **0.26** | **4.60** | **0.29** | **7.83** | **0.29** | **5.22** |
| Images | IL | 0.35 | 4.69 | 0.48 | 8.06 | 0.35 | 5.45 |
| | **Ours** | **0.27** | **4.30** | **0.31** | **7.67** | **0.18** | **4.93** |

Table 3: Real-world performance comparison between DAgger and our fine-tuned approach for two types of input representations.

a DAgger policy trained using the same number of 10M data samples. The results are detailed in Table 3, with supplementary videos providing visual footage. Our approach consistently achieved faster lap times and smaller gate errors in the real-world setting, confirming the effective real-world transfer of our vision-based quadrotor enhancements. Further analysis of the policies' real-world trajectories, depicted in Fig. 1 and Fig. 6, shows that fine-tuning results in tighter trajectories and higher peak velocities. Notably, in the challenging *SplitS* maneuver, the RL-fine-tuned policy executed tighter turns and longer straights, optimizing overall speed. These improvements illustrate that RL fine-tuning enables the discovery of maneuvers that enhance speed and performance, surpassing the capabilities of imitation learning alone.

## 5   Limitations and Discussions

In this work, we introduced a novel approach by fusing the strengths of Reinforcement Learning (RL) and Imitation Learning (IL) for vision-based agile quadrotor flight, specifically focusing on autonomous drone racing. We demonstrate for the first time a visuomotor policy capable of navigating through a sequence of gates using solely gate corners or RGB images. One limitation is that our current setup is tested in the controlled lab settings, it will likely fail in an in-the-wild setup. Despite demonstrating superior robustness compared to existing baselines, we believe the perception module in our framework is to improve to handle more out-of-distribution cases. In this work, we utilized a pre-trained ResNet to ensure a fair performance assessment. For future work, we aim to integrate a customized vision encoder that leverages data from diverse simulation settings, modalities, and extensive real-world environments.

**Acknowledgments**

This work was supported by the European Union's Horizon Europe Research and Innovation Programme under grant agreement No. 101120732 (AUTOASSESS) and the European Research Council (ERC) under grant agreement No. 864042 (AGILEFLIGHT). The authors thank Chunwei Xing and Ismail Geles for the insightful discussions.

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

# A   Supplementary Materials

## A.1   Quadrotor Dynamics for Policy Training

The quadrotor is assumed to be a 6 degree-of-freedom rigid body of mass $m$ and diagonal moment of inertia matrix $\boldsymbol{J} = \mathrm{diag}(J_x, J_y, J_z)$. Furthermore, the rotational speeds of the four propellers $\Omega_i$ are modeled as a first-order system with a time constant $k_{\mathrm{mot}}$ where the commanded motor speeds $\boldsymbol{\Omega}_{\mathrm{cmd}}$ are the input. World $\mathcal{W}$ and Body $\mathcal{B}$ frames are defined with an orthonormal basis i.e. $\{\boldsymbol{x}_{\mathcal{W}}, \boldsymbol{y}_{\mathcal{W}}, \boldsymbol{z}_{\mathcal{W}}\}$. The frame $\mathcal{B}$ is located at the center of mass of the quadrotor. The state space is thus 17-dimensional and its dynamics can be written as:

$$\dot{\boldsymbol{x}} = \begin{bmatrix} \dot{\boldsymbol{p}}_{\mathcal{WB}} \\ \dot{\boldsymbol{q}}_{\mathcal{WB}} \\ \dot{\boldsymbol{v}}_{\mathcal{W}} \\ \dot{\boldsymbol{\omega}}_{\mathcal{B}} \\ \dot{\boldsymbol{\Omega}} \end{bmatrix} = \begin{bmatrix} \boldsymbol{v}_{\mathcal{W}} \\ \boldsymbol{q}_{\mathcal{WB}} \cdot \begin{bmatrix} 0 & \boldsymbol{\omega}_{\mathcal{B}}/2 \end{bmatrix}^{\top} \\ \frac{1}{m} \left( \boldsymbol{q}_{\mathcal{WB}} \odot (\boldsymbol{f}_{\mathrm{prop}} + \boldsymbol{f}_{\mathrm{drag}}) \right) + \boldsymbol{g}_{\mathcal{W}} \\ \boldsymbol{J}^{-1} \left( \boldsymbol{\tau}_{\mathrm{prop}} - \boldsymbol{\omega}_{\mathcal{B}} \times \boldsymbol{J}\boldsymbol{\omega}_{\mathcal{B}} \right) \\ \frac{1}{k_{\mathrm{mot}}} \left( \boldsymbol{\Omega}_{\mathrm{cmd}} - \boldsymbol{\Omega} \right) \end{bmatrix} , \tag{3}$$

where $\boldsymbol{g}_{\mathcal{W}} = [0, 0, -9.81\,\mathrm{m/s^2}]^{\top}$ denotes earth's gravity, $\boldsymbol{f}_{\mathrm{prop}}, \boldsymbol{\tau}_{\mathrm{prop}}$ are the collective force and the torque produced by the propellers, and $\boldsymbol{f}_{\mathrm{drag}}$ is a linear drag term. The quantities are calculated as follows:

$$\boldsymbol{f}_{\mathrm{prop}} = \sum_i \boldsymbol{f}_i , \quad \boldsymbol{\tau}_{\mathrm{prop}} = \sum_i \boldsymbol{\tau}_i + \boldsymbol{r}_{\mathrm{P},i} \times \boldsymbol{f}_i , \tag{4}$$

$$\boldsymbol{f}_{\mathrm{drag}} = - \begin{bmatrix} k_{vx} v_{\mathcal{B},x} & k_{vy} v_{\mathcal{B},y} & k_{vz} v_{\mathcal{B},z} \end{bmatrix}^{\top} , \tag{5}$$

where $\boldsymbol{r}_{\mathrm{P},i}$ is the location of propeller $i$ expressed in the body frame , $\boldsymbol{f}_i, \boldsymbol{\tau}_i$ are the forces and torques generated by the $i$-th propeller, and $(k_{vx}, k_{vy}, k_{vz})$ [44, 45] are linear drag coefficients.

## A.2   Reward Formulations for RL Trainings.

The reward components are formulated as follows:

$$\begin{aligned} r_t^{\mathrm{prog}} &= \lambda_1 (d_{\mathrm{Gate}}(t-1) - d_{\mathrm{Gate}}(t)), \\ r_t^{\mathrm{perc}} &= \lambda_2 \exp(\lambda_3 \cdot \delta_{cam}^4), \\ r_t^{\mathrm{act}} &= -\lambda_3 \|a_t - a_{t-1}\|, \\ r_t^{\mathrm{br}} &= -\lambda_4 \|\boldsymbol{\omega}_t\|, \\ r_t^{\mathrm{pass}} &= c_1, \quad \text{if robot passes the next gate,} \\ r_t^{\mathrm{crash}} &= -c_2, \quad \text{if robot crashes (gates, ground) .} \end{aligned} \tag{6}$$

Here $d_{\mathrm{Gate}}(t)$ denotes the distance from the robot's center of mass to the center of the next gate to pass, $\delta_{\mathrm{cam}}$ is the angle between the camera's optical axis and the direction towards the center of the next gate. $\boldsymbol{a}$ represents the control command, and $\boldsymbol{\omega}$ the bodyrate. $\lambda_1, \lambda_2, \lambda_3, \lambda_4, c_1, c_2$ are different positive hyperparameters.

In our experiments, we employ identical hyperparameters for both state-based teacher training and vision-based RL fine-tuning to ensure a fair comparison. These parameters are determined based on iterations, shown in Table. 4 in both simulation and real-world experiments, aiming to achieve optimal and smooth performance for the state-based policy.

## A.3   Training Configurations.

For state-based teacher training, we employ a policy network consisting of a two-layer MLP, each layer containing 256 neurons, with a final layer outputting a 4-dimensional vector using a $tanh$ activation function. In imitation learning, a 3-layer Temporal Convolutional Network (TCN) is utilized to encode the 32 timestamps of perceptual inputs. The length of the temporal embedding is

| Reward Name | Symbol | Value |
|---|---|---|
| Progress reward | $\lambda_1$ | 0.5 |
| Perception-aware reward | $\lambda_2$ | 0.025 |
| Command smoothness reward | $\lambda_3$ | 2e-4 |
| Body rate penalty | $\lambda_4$ | 5e-4 |
| Gate passing reward | $c_1$ | 10 |
| Collision penalty | $c_2$ | 4 |

Table 4: Parameters for RL training.

128, followed by another two-layer MLP to output the control command. For imitation learning, we employ a batch size of 512, and convergence typically occurs after collecting 5M data samples over approximately 100 epochs. We incorporate a linear decay in the learning rate, starting at 1e-3 and decreasing to 1e-5 at 50 epochs, remaining unchanged for the remainder of the training process.

### A.4 Hardware Configurations.

We deploy our approach in the real world using a high-performance racing drone with a maximum thrust-to-weight ratio (TWR) of 5.78. However, for our experiments, we have limited the TWR to 2.7. We use a modification of the *Agilicious* platform [46] for the real-world deployment. We have replaced the onboard computer with an RF receiver, which is connected directly to the flight controller[1] and takes care of parsing the collective thrust and bodyrate commands from the computer. Additionally, we also mount an ultra-low latency camera feed sender, which sends the live video stream to the base computer. This configuration is similar to the one used by professional drone racing pilots.

### A.5 Ablation study on Asymmetric Critic Formulation

In stage III of our approach, the visuomotor policy undergoes fine-tuning using an asymmetric critic setup. In this experiment, we ablate how the critic configurations, as demonstrated in Fig. 3, can impact policy performance. As depicted in Fig. 7, RL fine-tuning with an asymmetric critic function achieves the highest reward within the same sample budget. At the same time, as shown previously, including privileged knowledge in the training process can also lead to better performance when handling partial observations.

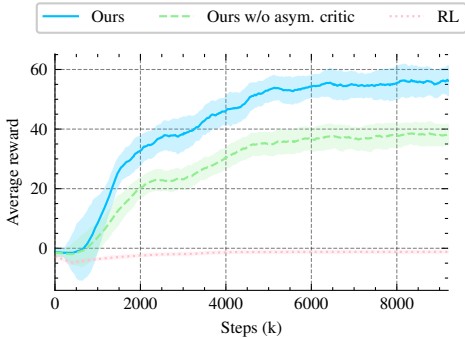

Figure 7: Comparing rewards across different RL configurations for the SplitS track using ResNet embedding, we find that utilizing an asymmetric critic makes the learning process more efficient. As a result, we have selected this configuration as the default setting for our other experiments.

### A.6 Performance w/ Diff. History Length

In Table. 5 we detailed the results of our approach against the best baseline methods DAgger on varying different history lengths. It is evident that by incorporating more historical information, the student could achieve a higher success rate. More importantly, in all of these cases, our approach achieves both better performance and success rate. The enhanced performance of our RL-based approach in partially observable situations can be attributed to our asymmetric actor-critic setup, where additional state information is provided for value estimations during environment interactions. This setup significantly mitigates the challenges of partial observability, thereby improving the robustness and effectiveness of the learning process.

---

[1] https://www.betaflight.com

| History | SR% | | MGE [m] | | LT [s] | |
|---|---|---|---|---|---|---|
| Length | DAgger | **Ours** | DAgger | **Ours** | DAgger | **Ours** |
| 4 | 0 | 0 | - | - | - | - |
| 8 | 28 | **84** | 0.64 | **0.29** | 8.34 | **7.92** |
| 16 | 58 | **97** | 0.52 | **0.27** | 8.31 | **7.83** |
| 32 | **100** | **100** | 0.27 | **0.22** | 8.26 | **7.68** |
| 64 | **100** | **100** | 0.28 | **0.21** | 8.27 | **7.65** |
| Average | 57 | **76** | 0.43 | **0.25** | 8.30 | **7.77** |

Table 5: Ablation study on history length of the policy observations using raw pixels. We could clearly find out by using more history observations, that the policy improvement will get improved. Notably, our approach consistently outperforms baseline methods across all history lengths

### A.7 Onboard Image Visualization

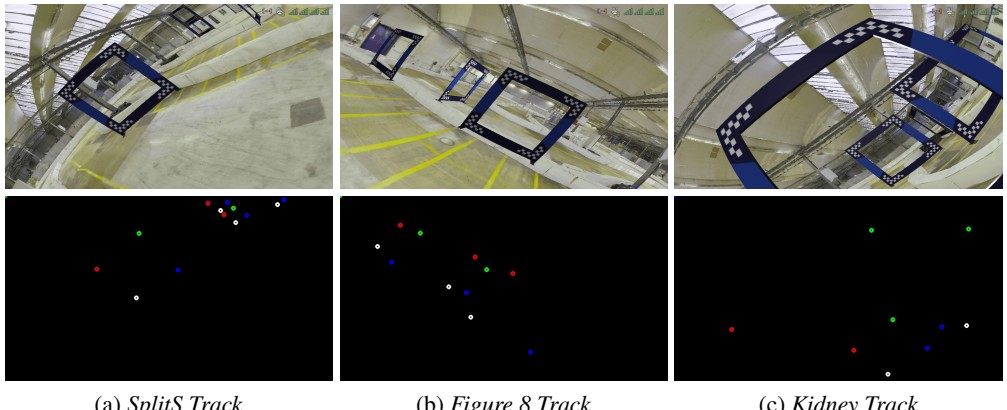

(a) *SplitS Track*      (b) *Figure 8 Track*      (c) *Kidney Track*

Figure 8: *Top:* Visualization demonstrates the render image input for our visuomotor policies in simulation. *Bottom:* Sparse corners visualization. Our learned visuomotor policies, relying solely on perceptual inputs, showcase their acquired capabilities for achieving robust but agile flying performance across three distinct tracks.

To significantly reduce the sim-to-real gap, we gather LiDAR and image data within our indoor testing arena and construct a digital twin for all our experiments. In Fig 8, visualizations of the images and the corners of our policies on three different racing tracks are depicted.

It is noteworthy that for corner generation, there is a 20% probability of missing data per corner, with ±10 pixels of noise applied. For a detailed view of real-world flights, please see the accompanying videos.

### A.8 Unobservable States Illustration

For imitation learning, the policy usually needs to infer action from only partially observable states, here we demonstrate one detailed example for corner-based racing in *SplitS Track* in Fig. 10. To avoid the fact that the policy will need to infer actions from unobservable states, we utilize the history of the observations and the asymmetric setup for RL training.

### A.9 Can we fine-tune an amateur policy into a champion-level policy?

In this experiment, we address the question of how far our setup can improve the performance of a policy learned through imitation learning. We adapted the parameters presented in [9] for this purpose. In this series of experiments, state information is used as input for all evaluations. First, we limit the maximum policy thrust to 6N per motor and train a teacher policy using PPO. Then, we train an imitation policy using DAg-

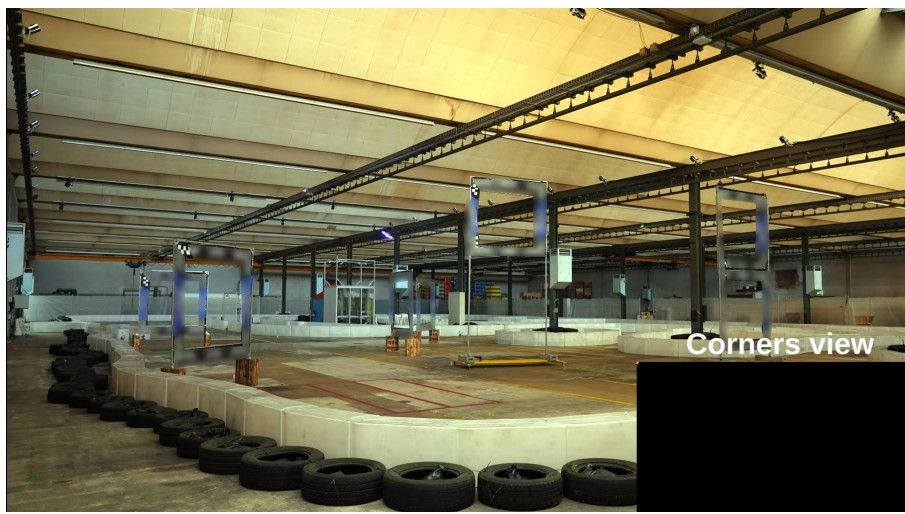

Figure 10: Illustration of one corner observation in the real world racing track. There are certain timesteps where there exist no meaningful corner projections at all. Hence we emphasize the necessity of introducing history information to handle unobservable perceptual states.

| Approach | Slow IL policy | | Our Finetuned Policy | | Champion-level Policy | |
|---|---|---|---|---|---|---|
| | LT [s] | SR [%] | LT [s] | SR [%] | LT [s] | SR [%] |
| Nominal Simulation | 9.53 | 39 | 5.17 | 100 | 5.14 | 100 |
| Realistic Simulation | 10.29 | 30 | 5.27 | 100 | 5.26 | 100 |

Table 6: Ablation study on history length of the policy observations using raw pixels. We could clearly find out by using more history observations, that the policy improvement will get improved. Notably, our approach consistently outperforms baseline methods across all history lengths

ger to imitate the slow policy, after which we apply our approach to fine-tune the policy with the full thrust of 11.3N per motor. The detailed training curve is visualized in 9.

We demonstrate that our policy training achieves improved sample efficiency, even for the state-based approach. The quantitative results, shown in 6, clearly indicate that our approach greatly improves policy performance, achieving lap times within a difference of 0.05s to that in [9], where they outperformed human champions.

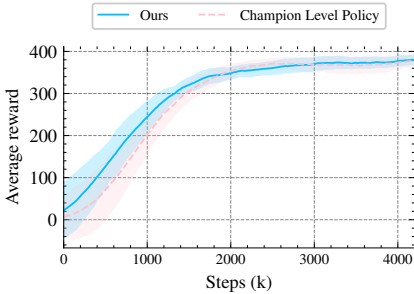

Figure 9: Return comparison between our approach and the policy achieved Champion-level performance.

