# OpenReview forum: "Bootstrapping Reinforcement Learning with Imitation for Vision-Based Agile Flight"
_robot-learning.org/CoRL/2024/Conference — CoRL 2024_

### Official Review · Reviewer_8K8j · 2024-07-12
**Simple and powerful technique but missing baselines**

**Originality:** 3
**Technical Quality:** 3
**Clarity Of Presentation:** 3
**Potential Impact:** 3
**Recommendation:** 3
**Confidence:** 3

**Review:**

**Quality:** The approach is sensible, well-motivated and demonstrates strong results over baselines. However, I do feel as if there are some obvious shortcomings in the results and analysis: for baselines, the RL baseline is from scratch, but another obvious approach would be to mix the privileged state information used by the authors approach, with the image inputs. Since this is not the first paper to use privileged information, it would've been nice to see results against other existing methods. In terms of analysis, it would be insightful to understand the importance of the fine-tuning stage (is this the Dagger baseline? it is unclear).

**Clarity:** The paper is mostly well-written. There are some elements of the experimental results that could be more clear--for example it is unclear how the Dagger baseline is performed.

**Originality & Significance:** While the method itself offers mild novelty over existing approaches, there is originality and significance in the system design, particularly for the area of quadrotors.

**Quality Of The Limitations Section:**

3

**Questions For Rebuttal:**

- Include a RL baseline which uses privileged state information.
- Can you explain the Dagger baseline in more detail? How does this differ from your approach without the fine-tuning stage?

**Robotics Focus:**

4

**Summary Of Paper:**

The authors introduce a method for training a visuomotor policy by a three-stage training procedure: RL with privileged state information, IL, fine-tuning. The effectiveness of their approach is demonstrated in simulation and with a real-world quadrotor.

**Summary Of Recommendation:**

Straightforward method with convincing results in a real-world system.

---

### Official Review · Reviewer_e7LB · 2024-07-20

**Originality:** 2
**Technical Quality:** 4
**Clarity Of Presentation:** 4
**Potential Impact:** 2
**Recommendation:** 3
**Confidence:** 3

**Review:**

The paper is clearly well written with high-quality images illustrating the results and proposed method. The problem setting is definitely original and the results are very well done. The experimental results seems very exciting and demonstrate what was done to be effective. In terms of strength, the paper does provide good motivation for the work and the missing components: sample-efficiency and mitigating catastrophic forgetting.

The main weakness of the work is that results do not necessarily support the sample-efficiency claim nor does the approach have any certainty that catastrophic forgetting is mitigated at all. Though there are results in the supplementary, it seems it is just DAgger with an ablation study on how much data it useful and a low-learning rate to prevent catastrophic forgetting. Specifically, at the start of page 5, the claim that the policy step should be kept small, while it may work for this specific problem, seems arbitrary and difficult to reproduce. What exact values are good learning rates versus bad and what can be learned from the appropriate learning rate?

**Quality Of The Limitations Section:**

3

**Questions For Rebuttal:**

First, the paper should address the following on sample-efficiency novelty claim:

How is the approach improving sample-efficiency?
Is there a specific gain in certain training components that do enhance sample-efficiency more than other components?

Second, regarding mitigation of catastrophic forgetting claims:

At what step-size does catastrophic forgetting not occur?
Are there any specific guarantees or rule of thumb based on empirical analysis that can be used or followed?

Third, regarding results:
In table 2, what happens at 10 % disturbance?
Does the system perform just as good or does performance degrade significantly?
Additional results on catastrophic forgetting would improve the paper's reproducibility.

**Robotics Focus:**

4

**Summary Of Paper:**

This paper presents a method for optimizing vision-based flight controllers via reinforcement learning with imitation learning as an initialization.

**Summary Of Recommendation:**

While the results and platform are exciting, the contributions currently stated are not strongly supported and come across as application of existing methods. More work should be done to highlight the contributions and support the claims.

---

### Official Review · Reviewer_tjv6 · 2024-07-21
**The paper makes advancements towards true end-to-end agile robot control.**

**Originality:** 3
**Technical Quality:** 4
**Clarity Of Presentation:** 4
**Potential Impact:** 3
**Recommendation:** 3
**Confidence:** 4

**Review:**

The paper is very well written, easy to follow, and supported by good/clear figures and tables. (The only suggestions I would have in terms of presentation is to explain all acronyms used in the tables also in their captions and fix the fact that some of the paragraph titles have title case, some have sentence case.) The experiment results section is, in a few instances, a bit less clear and I added a few questions for clarification below. The methodology is sufficiently clearly explained and multiple options/ablations (with regard to, race tracks, visual encoding and the imitation learning strategy) are explored. The paper is indeed very relevant to robotics and autonomy and, in a broader sense, could also serve as a starting point to better understand how humans implicitly understand and perform complex physical tasks almost only relying on vision and few other propioception/auditive cues. In this sense, the main thing that is missing from the paper is a direct comparison with a human and/or the other learning and state-based approaches that have achieved super-human performance in this task.

**Quality Of The Limitations Section:**

3

**Questions For Rebuttal:**

- "we validated our policy in real-world scenarios using Hardware-in-the-Loop (HIL) simulations, aided by a VICON motion capture system to generate perceptual inputs through rendering or gate corner projections." this sentence is ambiguous, how does the mocap system aid the rendering?
- "It is important to note that both i) and ii) are not domain randomization during the training" meaning
- why imitate only from a state-based RL policy and not also from expert-collected data?

**Robotics Focus:**

4

**Summary Of Paper:**

The paper presents how to combine reinforcement and imitation learning to distill a learning-based control policy to autonomously complete a drone racing track solely from vision inputs.

**Summary Of Recommendation:**

The paper is a pleasant read and an interesting stepping stone in autonomous aerial robotics.

---

### Author Rebuttal · Authors · 2024-08-09

We thank the reviewer for carefully reading the paper and providing valuable feedback. The reviewers have stated that our paper is “very well written”, the method is "simple and powerful", the experimental results are “exciting”, and the problem is “well motivated”.

Based on the initial reviews and suggestions, we added some extra experiments and revised our manuscripts correspondingly. The changes (marked in blue text) to the manuscript are summarized below.

- We revised several unclarified acronyms and changed the title of the subsections to title cases.
- We revised some unclear sentences in the experiment section that could cause confusion or ambiguity.
- In the Methodology section, we included more explanation of Algorithm 1.
- In Experiment *Robutstness to unknown perceptual and dynamical disturbances* , we added the experiments by comparing our approach with the IL baseline with 10% disturbances; the results are in Table 2.
- In Experiment **Section 4.1** and Supplementary **A.14**, we included the detailed training parameters and the training loss visualization for our student policy training.
- In Supplementary **A.12**, we added the hyperparameters and the learning rate visualization for our adaptive learning approach.
- In Supplementary **A.13**, we added the trajectory and action output comparison of the behavior cloning policy and our approach.
- In Supplementary **A.16**, we added the comparison of our vision-based policy against a state-based policy.

We thank the reviewers again for the valuable feedback and are happy to discuss further during the rebuttal period.

---

### Decision · Program_Chairs · 2024-09-04

**Decision:**

Accept

**Comment:**

Strengths: The paper is very well written, easy to understand and well motivated.

Weaknesses: Additional comparisons to relevant baselines and a better substantiation for both sample-efficiency novelty and mitigation of catastrophic forgetting are needed.

-----

The effort the authors put into the careful rebuttal is commendable and in the revised paper the strengths clearly outweigh the weaknesses.